# Intrinsic Physical Unclonable Function (PUF) Sensors in Commodity Devices

**DOI:** 10.3390/s19112428

**Published:** 2019-05-28

**Authors:** Shuai Chen, Bing Li, Yuan Cao

**Affiliations:** 1Shenzhen Research Institute, SEU-FiberHome Joint Research Center, School of Cyber Science and Engineering, School of Microelectronics, Southeast University, Nanjing 210000, China; chenshuai_ic@seu.edu.cn (S.C.); bernie_seu@seu.edu.cn (B.L.); 2College of Internet of Things Engineering, Hohai University, Changzhou 213000, China

**Keywords:** physical unclonable functions, PUF sensor, DRAM PUF, key generation scheme

## Abstract

The environment-dependent feature of physical unclonable functions (PUFs) is capable of sensing environment changes. This paper presents an analysis and categorization of a variety of PUF sensors. Prior works have demonstrated that PUFs can be used as sensors while providing a security authentication assurance. However, most of the PUF sensors need a dedicated circuit. It can be difficult to implemented in commercial off-the-shelf devices. This paper focuses on the intrinsic Dynamic Random Access Memory (DRAM) PUF-based sensors, which requires no modifications for hardware. The preliminary experimental results on Raspberry Pi have demonstrated the feasibility of our design. Furthermore, we configured the DRAM PUF-based sensor in a DRAM PUF-based key generation scheme which improves the practicability of the design.

## 1. Introduction

In the internet of things (IoT) era, billions of smart devices are connected and interact with each other. A large number of sensor nodes are distributed in the network for sensing the world. The data collected from the sensors are used to trigger the whole system to interact. However, the sensing, collecting and communication of sensor data are vulnerable to attacks [1].

The sensors are working in a challenging world. This variable and hash environment results in the sensors being prone to failure. Furthermore, the sensors that are distributed all over the world can be physically accessed by the attacker. Therefore, they are also vulnerable to physical attacks. For example, some physical attacks focus on the long-term private key stored in non-volatile memory (NVM) that is assumed to be secure. The secret data stored in NVM like Read-Only memory (ROM), Electrically Erasable Programmable Read-Only Memory (EEPROM) and flash can be recovered even after erasures [2]. Non-invasive, semi-invasive and invasive attacks [3] can extract the private key, making NVM the weak link in many security implementations.

Furthermore, the sensor nodes are limited in computation, memory and power because of the resource constraints. Therefore, certain traditional security solutions cannot be embedded in it. Also, there are billions of connected devices with different manufacturers and service providers. Thus the nodes may not have global identifications. Therefore, it is hard to authenticate the identity of each node to countermeasure the false ones [1].

Another emerging threat for sensor nodes is sensor spoofing attacks [4]. The attackers can spoof a false analog signal to the sensor which may cause malfunctions. It is hard to address this attack because sensors cannot inherently distinguish between malicious and non-malicious signals. One promising solution is the so-called sensor fusion [4]. By comparing the sensed data from various sensors, the malicious signal can be detected. However, for some low-cost sensor nodes, the overhead of multi-sensors is unacceptable. Therefore, the low-cost intrinsic sensors that do not use an analog-mixed circuits are worth studying.

Physical unclonable functions (PUFs) [5] exploit the random variability of nano-scaled manufacturing variations to achieve tamper resistance, and physical and mathematical unclonablility. It has become an indispensable primitive to countermeasure the aforementioned security issues, since:PUF provides the possibility for low-cost key storage and authentication which is not vulnerable to physical attacks [6,7]. No secret data needs to be stored in NVM. Instead, the secret key or identification is derived from the physical properties of the PUF when needed.Some PUFs are very sensitive to environmental parameter changes. Therefore, one can use the PUF response to sense environmental changes. For example, a ring oscillator (RO) PUF can measure the temperature in the Field-Programmable Gate Array (FPGA) boards [8]. The digital PUF is easily embedded in the Application Specific Integrated Circuit (ASIC) or FPGA without analog circuits. Therefore, a PUF-based sensor is a good candidate to countermeasure a spoofing attack where the outputs from multiple traditional and PUF sensors can be compared to catch an anomaly [9].Some digital PUFs can be used as a fusion of low-cost key storage, authentication and sensor. It is easy to be implemented in digital devices because it does not require any analog-mixed process.

However, most of the existing PUF sensors rely on dedicated circuits that are very difficult, if not impossible, to find in off-the-shelf commodity devices [8,9,10]. The addition sensors may not meet the requirements of some low-cost systems. Therefore, some intrinsic PUF instances within standard hardware that do not need any dedicated circuits or hardware modifications can be evaluated as PUF sensors to overcome the requirements of the off-the-shelf devices. In this paper, we propose a temperature sensor leveraging Dynamic Random Access Memory (DRAM) PUF in commodity devices. Our method is based on the existing DRAM circuits and does not require adding any hardware circuits in the device.

Our key contributions are as follows:Implementation of an intrinsic DRAM PUF-based temperature sensor in off-the-shelf commodity devices.Test the feasibility of the DRAM PUF-based sensor and configure it in a DRAM PUF-based key generation scheme.

The rest of the paper is organized as follows. Background and some related security issues are introduced in Section 2. In Section 3, we summarize the existing PUF sensors. A novel temperature sensor based on DRAM PUF is proposed in Section 4. Evaluation of the novel temperature PUF is presented in Section 5 and we discuss this work in Section 6. The conclusion is presented in Section 7.

## 2. Background

### 2.1. Physical Unclonable Functions

Since the introduction of an optical PUF in 2002 [11], researchers have proposed various PUF designs. The digital PUFs are the most popular components. The essence of a digital PUF is a hardware circuit with unique binary or analog behavior which depends on the integrated circuit (IC) manufacturing variations, e.g., delays, frequencies or capacitances. The process variations are randomness, even the manufacturer can not predict or clone it. Hence, PUFs have been proposed as an important building block for security systems. For example, PUFs can be used in a lightweight key storage scheme [6,12] or authentication and identification scheme [13,14] which does not need any NVM to store the secret data. The private key is derived from the PUFs during run time instead of being stored in the NVM. Thus, it can be used to protect against certain NVM attcks.

Most of the digital PUF designs (e.g., Arbiter PUF [15] and Ring Osillator (RO)-PUF [16]) require the design of dedicated circuits which tend to be rather complex in design and manufacturing. Also, it is difficult, if not impossible, to find these dedicated circuits in existing commodity devices. Therefore, people are becoming more and more interested in intrinsic PUFs, e.g., Static Random-Access Memory (SRAM) PUF [17] and DRAM PUF [18]. DRAM PUF is the focus of this work.

### 2.2. Secure Key Management Scheme for Sensor Networks

Key management is considered the most critical component of security systems [19], as the leakage of keys makes even the toughest cryptographic system pliable. It is the same in the trust management scheme in the wireless sensor network (WSN) [20,21]. To maintain the tolerant level of trust among the sensor nodes, trust management is established to authenticate the genuine and fake sensor nodes. In the trust management system, a robust and lightweight key management scheme is critical.

In paper [22], the author divided the key management schemes into symmetric, asymmetric and hybrid, based on the encryption techniques. For the scenario of dynamic WSN, paper [23] proposed a dynamic key management scheme by refreshing the pairwise keys periodically or on demand. There is also an existing survey on key management in WSN that classifies the key management schemes as key pre-distribution schemes, hybrid cryptography schemes, one-way hash schemes, key infection schemes, and key management in hierarchy networks. In the key pre-distribution schemes, one lightweight solution is that all the nodes only need to store a master secret key. When used in the field, the key management scheme is initiated by the global master key. However, due to the NVM-based key storage scheme, the previous works showed that any micro-controller, FPGA, secure memory, smart-card and even ASIC can be attacked successfully by several attack methods [1,24,25]. The whole WSN will be compromised if one sensor node key is promised.

One complement to the aforementioned leakage attacks is proposed in paper [26]. The authors proposed a public-key encryption scheme that is resistant to key leakage attack. However, the public-key based crypto desigh is too expensive to implement in the low-cost systems.

A natural defense would be to store the secret key in tamper-resistant hardware. For example, paper [27] proposed a countermeasure using the coating layer and paper [28] presents a construction by the error detection codes that is resilient to key leakage. However, the traditional tamper-resistant hardware might also vulnerable to some attacks, e.g., [25,29]. Furthermore, it is difficult to implement in the resource-constrained sensor nodes and the off-the-shelf devices.

PUF provides the possibility for a tamper-resistant, low-cost key storage and authentication scheme [7,30]. The advantages of this combination would be:Tamper-resistant. The key is extracted from the nano-scale manufacturing variations, not “burned” in the NVM like EEPROM. Therefore, even an invasive attack cannot compromise the secret key.Low-cost. For some intrinsic PUFs, the security system does not need to add any dedicated circuit in hardware. For example, the implementation of DRAM PUF [31] just requires the firmware modifications.Combing node identity. Integrating node identity in the process of key production will make a system more secure [32]. It is also helpful for the resistance of node replication attacks [33]. PUFs can be seen as the “fingerprints” of hardware, it can be used in identification and key generation. In the PUF-based key generation scheme, the key is extracted from the hardware feature. Obviously, this feature can be seen as the identity of nodes.

## 3. PUF Sensor

Based on the roles of PUF in PUF sensors, we classify the current PUF sensors as PUF-protected sensors and PUF-based sensors.

### 3.1. PUF-Protected Sensor

The so-called PUF-protected sensor is a variety of sensor that leverages the functionality of a conventional PUF to authenticate the sensor nodes or protect the sensed value.

The PUF-protected sensor was first proposed in paper [34]. As shown in Figure 1, the conventional PUF is co-mingled with the sensor so that the sensor value is determined by both the physical quantity and PUF response. In paper [34], the signals of the offset generator are selected randomly by the PUF response to generate the final sensed value. Like the PUF-based authentication process, the proposal also has an enrollment phase to store some challenge-measurement-response pairs. When used in the field, the micro-controller just accepts the sensed data that passed the verification process. Therefore, the PUF-protected sensor becomes a promising mechanism for securing remote sensors.

Cao et al. extracted some PUF features from the CMOS image sensor [35] to improve the image sensor as a trusted entity. Each pixel can generate a 1-bit PUF response based on the fixed pattern noise resulting from manufacturing variations. Therefore, each image sensor can generate a unique and reliable signature for the pictures using the hash function. This design can be implemented on the existing CMOS image sensors without a dedicated circuit. It can be used in a PUF-based perceptual image hash scheme to carry out the image content birth certification.

### 3.2. PUF-Based Sensor

As shown in Figure 2, the PUF-based sensors evaluate the environmental parameters based on the environment-dependent-behavior of the PUF. Usually, the PUF-based authentication and key generation scheme has two steps: The enrollment phase in the security environment, and the authentication or key generation phase when used in the field. Most PUFs exhibit unreliability problems due to inherent sensitivity to the environmental conditions, e.g., temperature and supply voltage [36]. This unwanted fact gives us a new idea to sense environmental changes.

Paper [37] presented a micro-electro-mechanical (MEM) relay based RO-PUF to sense pressure and provide authenticity. Compared to the conventional RO-PUF, the “sensorPUF” leveraged the MEM relay inverter to replace the CMOS inverter. The MEM relay inverter can sense the pressure changes and influence the behavior of RO-PUF to a unique but deterministic function. Therefore, the sensed value has both the pressure feature and the hardware “fingerprints” feature to realize an authenticated pressure measurement. However, it is hard to implement this design in some low-cost IoT devices.

Compared to the aforementioned work, paper [10] proposed a universal RO-PUF to sense voltage. By leveraging the sensitivity challenges, the authors also investigated a challenge selecting a method to improve the sensing capability. Similarly, paper [9] presented another type of voltage sensor based on the error rate or universal Glitch PUF.

Furthermore, the oscillation frequency is sensitive to temperature changes [8,38]. It can be used as the thermal sensor in FPGA to monitor the die temperature. This temperature sensor also can be used as a possible malicious application of the thermal covert channel. The transmitter can encode the transmitted data into heat patterns and the RO-PUF based temperature sensor can detect temperature changes in the receiver [39,40].

## 4. Proposed Intrinsic PUF Sensor Based on DRAM PUF

Although the aforementioned design allows the sensor to inherently provide assurance of authenticity by co-mingling sensing and unique hardware features, it is difficult, if not impossible, to be found in the existing off-the-shelf commodity devices. Therefore, the design and implementation of intrinsic PUF sensors that do not need to add any dedicated hardware are necessary. This paper briefly presents a novel intrinsic temperature sensor based on the decay feature of DRAM PUF.

### 4.1. DRAM PUF

DRAM is pervasively used in existing embedded systems. As shown in Figure 3, the DRAM cell consists of a transistor and a capacitor. Each cell stores 1-bit data in the capacitor and can be accessed through the transistor. The cells are grounded in a 2-dimensional array, where each row is connected to a word line and column linked in a bit line. In each cell the capacitor leaks the charge over time which causes data to flip from the previous contents. Therefore, DRAM chips usually have a periodical self-refresh module to recharge the capacitor on time which is controlled by the memory controller.

Due to the manufacturing variations among DRAM cells, some cells leak faster than others. After a certain delay time, enough charge has leaked crossing the threshold from some cells such that the stored logical bit flips. For the other cells, the contents stay stable. This behavior heavily depends on the random manufacturing variations and environment (e.g., temperature). The random data flips allow DRAM to be a good candidate for PUF [31]. It can be used as a run-time accessible DRAM PUF in the key generation or authentication scheme in commodity devices [18]. Beyond that, Tehranipoor et al. [41] attempted to use the random start-up value of DRAM as a PUF. However, this method needs to control the power supply of the DRAM chip like the SRAM PUF. It is difficult to be implement in commodity off-the-shelf devices. In [42], the authors introduced error patterns bound to manufacturing variations of DRAM by reducing DRAM read access latency below the minimum value in data-sheet specifications to implement DRAM PUF. In this scenario, the computer system needs at least two DRAM ranks because normal read latency should be maintained at least in one rank to keep systems operational. Therefore, this method is difficult to implement in low-cost embedded systems. For example, we tried to implement this proposal in Raspberry Pi (Rpi) B+. However, Rpi B+ just have one rank. The system crashed when we set the value of reading latency as less than 2. In our study, we use the decay-based DRAM PUF as the covert channel to sense temperature changes in the commodity off-the-shelf devices.

### 4.2. Implementation of DRAM PUF on Raspberry Pi B+

We implemented and tested our sensor on three Rpi B+ development boards which are the most popular commodity embedded platform. Each board have a Broadcom BCM 2835 systems on chip (SOC) module which includes a 700 MHZ ARM11 76JZF-S processor and a VideoCore IV that implements a 512 MB Double Data Rate SDRAM (DDR2) memory.

We modified an open source firmware of Rpi [43] in order to get the privilege of DRAM decay control. The refresh of the whole DRAM has to be disabled as we can not control part of the DRAM address on Rpi. Similar to the previous work in the paper [31], we set the selective refresh by loops over all memory address that need to be refreshed by issuing a read to the first word in every DRAM row. Therefore, during query process of DRAM PUF, the other applications can operate normally.

Figure 4 shows the structure of DRAM PUF on Rpi B+. There are three important parameters for DRAM PUF: PUF address, initial value and decay time. PUF address is the DRAM address that supposed to be used as PUF. The initial value is a set of digital data that used to initiate the PUF address before the PUF query. In the PUF query, decay time indicates how long the DRAM PUF is disable refreshed. In our implementation, we set a 16 MB DRAM PUF with initial value = 1 and decay time = 60 s. These parameters can be compiled in the kernal of Rpi B+ or acquired from the upper computer via Universal Asynchronous Receiver/Transmitter (UART). And then, the PUF query code running on Graphics Processing Unit (GPU) can gain these parameters from Central Processing Unit (CPU) by the mailbox. The programs running on the CPU and GPU are only able to communicate via the mailboxes. All the aforementioned work has been published on the paper [44]. In paper [44], the decay feature of DRAM was used as covert channel leveraging the PoP architecture.

### 4.3. Embedded in the PUFs-Based Key Generation Scheme

The DRAM PUF-based key generation scheme has two phases: enrollment and key generation. In the enrollment phase, the security system generates helper data h=GEN(r), where GEN() is the generation process of helper data algorithm [45] and *r* is the output of DRAM PUF. *h* can be generated and stored in the devices or the data center. In the key generation process, when used in the field, the device queries the DRAM PUF and receives a measurement $r^{\prime }$. By the helper data algorithm, the system can regenerate the original DRAM output $r=REGEN(R^{\prime },h)$ if the Hamming Distance of *r* and $r^{\prime }$ is smaller than the error-correction capability of helper data algorithm, where REGEN() is the error correction process of the helper data algorithm. In the DRAM PUF-based sensor, the Hamming Distance of $r^{\prime }$ and *r* is the number of bit flips caused by the temperature changes. Therefore, as shown in Figure 5, the cross-correlation between the errors in DRAM PUF-based key generation process and the temperature variations is established. The novel temperature sensor is integrated into the conventional PUF-based key generation scheme.

For DRAM PUF, one promising key generation scheme was proposed in paper [31]. The authors discretize the decay feature by divided the DRAM cells into fast cell, slow cell and no cell. The decay time of fast cells is shorter than slow cells. Some randomly selected fast cells and slow cells, which are either extremely fast or extremely slow, are used to generate a key. In our tests, we simplified the implementation of this scheme by ordering these selected cells in their physical address. The contribution of this scheme is that the output of this proposal can be capable of higher randomness with higher reliability. The reliability of this scheme has been tested in paper [31].

## 5. Experimental Set Up and Evaluation

### 5.1. Experimental Set Up

Figure 6 presents the schematic of the experimental set up used to verify the feasibility of our design. It includes a thermal chamber with a thermal chamber controller; three Rpi B+ boards running the modified open source firmware that can communicate with the workstation via UART; and a workstation running the control scripts. The automatic test process is the following:The workstation sets the temperature of thermal chamber by the thermal chamber controller and starts the loop to monitor the temperature.When the temperature requirements are reached, the workstation writes the parameters of the DRAM PUF to the CPU.Execute the DRAM PUF query process on GPU and count the number of bit flips.Restart from step 1 for next set of parameters.

### 5.2. Test Results

We measured the temperature sensor instances on three Rpi B+ boards (Rpi1, Rpi2 and Rpi3) with temperature t= 15 ${}^{\circ }$C, 20 ${}^{\circ }$C, 25 ${}^{\circ }$C, 30 ${}^{\circ }$C, 35 ${}^{\circ }$C, 40 ${}^{\circ }$C and decay time 60 s. In each device, we measured one 16 MB DRAM in stride in the free address area of DRAM.

Decay time and temperature are two parameters that affect the number of bit flip for DRAM PUF. In our proposal, we evaluate the temperature, leveraging the number of bit flips of DRAM PUF. In Figure 7, we show the dependency between temperature and number of bit flips under certain decay times. The temperature changing was achieved using a thermal chamber. Although decay time affects the number of bit flips significantly, it does not influence the dependency between temperature and decay time. The temperature characteristics of DRAM PUF are very similar for different decay times. In the following tests, we evaluate our DRAM PUF under 30 s.

Figure 8 shows the number of bit flips of each Rpi, environment temperature and SoC temperature. Every point in the plot represents a test result under one temperature conditions. We see that the number of bit flips significantly increases with the temperature rising. However, there are obvious differences in the change of slope rate of the environmental temperature and number of bit flips among all the devices. The curve slope of temperature is very stable compared with the number of bit flips.

Prior work has presented that the decay time (retention time) of DRAM cells decreases exponentially as the temperature increases [46]. In paper [31], the authors computed it by the formula ${t^{\prime }}_{T^{\prime }}=t\cdot e^{-\alpha (T^{\prime }-T)}$. At temperature $T^{\prime }T$, DRAM PUF can generate a similar response under decay time $t^{\prime }t$. Furthermore, the number of bit flips is determined by the decay time under a certain temperature. Therefore, we analyse the relationship between temperature and number of bit flips by a simple fitting formula exponentially. The behaviour of bit flips $N_{T^{\prime }}$ under temperature $T^{\prime }$ can be computed by known parameters $N_{T}$ and *T* by:(1)$$N_{T^{\prime }}=N_{T}\cdot e^{\alpha (T^{\prime }-T)}$$

Based on our measurements, we estimated α to be 0.2465 for Rpi1, 0.2432 for Rpi2 and 0.2659 for Rpi3. As shown in Figure 9, the smooth fitting curve coincides very well with the original line with “X” label.

As shown in Table 1, the accuracy of our DRAM PUFs based temperature sensor can be within 4 ${}^{\circ }$C. Due to the reason that the accuracy of our thermal chamber can only be stabilized at 4 ${}^{\circ }$C to 5 ${}^{\circ }$C, we implemented our tests under that range. However, we can still find the big gap between our test results, e.g., from 20 ${}^{\circ }$C to 25 ${}^{\circ }$C, the number of bit flips increased threefold. Therefore, the real accuracy of our proposal should be much better than our test results. Theoretically, if the number of bit flips caused by noise is 200 (this will be shown in the following contents), the temperature accuracy should be less than $3{\;}^{\circ }$C ( $T^{\prime }-T=(ln\left(\left(188+200\right)÷188\right)÷0.2465$).

In our proposal, the uniqueness of DRAM PUFs is an important parameter to make sure that the DRAM PUF can generate a unique ID and key for the sensor. However, the uniqueness evaluation method based on inter-Hamming distance is not suited for DRAM PUF, because the majority cells of DRAM PUF does not flip in short decay time. Therefore, paper [31] proposed to use the Jaccard index to evaluate the uniqueness of multiple DRAM PUF instances. For two sets *A* and *B*, the Jaccard index is defined as Equation (Equation 2). As shown in Figure 10, the inter-chip Jaccard index is $1.4981\times 10^{-4}$. This small value indicates the high uniqueness of our DRAM PUF on Rpi.
(2)$$J\left(A,B\right)=\frac{A\cap B}{A\cup B}$$

For the DRAM PUF-based key generation scheme in Section 4.3, we tested the randomness of the output. As shown in Table 2, the scheme can generate random bits under 40 ${}^{\circ }$C, 30 s or under low temperature with longer decay time calculated by the aforementioned equation ${t^{\prime }}_{T^{\prime }}=t\cdot e^{-\alpha (T^{\prime }-T)}$.

Furthermore, to evaluate the robustness of our proposal, we tested the aging and workload effects of DRAM PUF and analyzed the voltage effects by the related study.

All the accelerated aging experiments are performed using a thermal chamber. Theoretically, one day’s test under $80{\;}^{\circ }$C is equal to 18 months of operation under room temperature [47]. Therefore, as shown in Figure 11, after 15 days accelerated aging tests, we can evaluate nearly 270 months of aging effects of DRAM PUFs. Although there are some fluctuations, the test results do not present serious aging effects. Similar conclusions are also shown in paper [47].

As shown in paper [48], voltage is one of the parameters that effect the junction leakage current $I_{leak}\propto e^{V_{appliedvoltage}}$. The relationship between the retention time of DRAM and $I_{leak}$ can be expressed as $T_{ret}\propto C_{s}/I_{leak}$, where $C_{s}$ is a parameter. Thus, as the increase of voltage, the retention time of DRAM decreased. For a certain decay time, there would be more bit flips. This may effect the robustness of DRAM based key generation scheme and the accuracy of the temperature sensor. Therefore, a stable supply voltage for DRAM chip is necessary for our proposal.

Furthermore, the working conditions of SoC may also influence the feature of our design. However, until now, the open source firmware for Raspberry Pi B+ cannot boot up a real operation system. Therefore, in this paper, we only tested the effects of two specific functions that operated on GPU. The difference between the two functions is reliable at 5%. These orders of magnitude are much smaller than the influence of temperature (as shown in Table 1). On the other hand, the SoC can mitigate the influence of workload by control the code operation when used in the field.

## 6. Discussion

### 6.1. Security Discussion

Since our intrinsic DRAM PUF relies on the intrinsic DRAM memory, memory protection is the premise for the security of our DRAM PUF-based key storage and sensor scheme. The memory used as PUF must be protected from tampering from all software outside the trust boundary. Furthermore, the privilege to control the register about DRAM decay control is another important security issue. The security system must make sure that only the legal component that is granted the highest privilege can access arbitrary memory without any limitations.

Isolation techniques for multi-core platforms that are based on resource partitioning offer a promising method for this security issue. For example, memory isolation techniques can control memory access rights on an embedded system. It relies on temporal partitioning of memory between trusted and untrusted code to create isolated memory for the execution of sensitive code. It can be used to protect a specified memory space by forbidding any illegal access to the memory address. The isolation techniques have been widely used in the existing computer security scheme, e.g., Intel Trusted Execution Technology (TXT) [49], Intel Software Guard Extensions (SGX) [50] and ARM TrustZone [51].

Although the memory resources that are used for DRAM PUF can be protected by the isolation technology aforementioned, another possible threat for our DRAM PUF-based sensor is the Rowhammer attack [52]. As shown in Figure 12, the attacker could try to introduce some random errors (bit flips) into DRAM PUF by repeatedly accessing adjacent rows which are legal for the attacker. If the DRAM PUFs address is discrete physically, this attack should be very powerful because there will be a lot of adjacent rows that can be used to operate the attack. On the contrary, if the DRAM PUF is physically successive as shown in Figure 12, the attack would be not very useful in our design because the rowhammer attack can only injure the borders of the DRAM PUF area. The number of bit flips introduced by this attack is very limited. Furthermore, two “empty-rows” can be used to isolate the DRAM PUF from the attacker.

### 6.2. Future Works

This novel DRAM PUF-based sensor still leaves a number of open research issues and questions that need to be addressed. The possible future work includes:We only verified the feasibility of the DRAM PUF-based sensor. Therefore, more comprehensive tests are necessary.The open source firmware [43] used in our implementation cannot boot up a whole operating system now. Therefore, we still are not clear about the influence of the operating code on the feature of the sensor.The query process of DRAM PUF needs several seconds of decay time. Therefore, it can not be used in certain real-time scenarios. Our future work will utilize more intrinsic PUF designs to address this issue.

## 7. Conclusions

In this work, we presented an analysis and categorization of the security key management schemes for sensor networks and PUF sensor designs. Previous works depicted that PUF can provide low-cost key storage and intrinsic sensors to countermeasure the security issues of physical attacks for NVM-based key storage and spoofing attacks. However, the existing PUF sensors can not be used in commodity off-the-shelf devices because of the dedicated circuits of PUF implementation. Our work demonstrates that intrinsic PUFs can be a good candidate to configure the PUF-based key storage and PUF sensor in the commodity off-the-shelf devices without any hardware changes. An evaluation of the DRAM PUF found on the off-the-shelf commodity device–Rpi B+, showed the feasibility of a DRAM PUF-based temperature sensor. Moreover, we proposed a DRAM PUF-based key storage scheme that can configure the PUF sensor in it. The sensor process can be operated during the key generation process. 

## Figures and Tables

**Figure 1 sensors-19-02428-f001:**
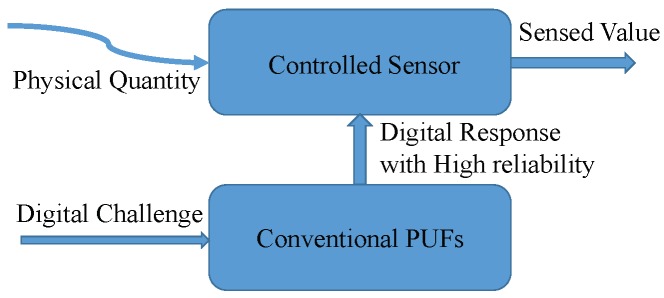
Physical unclonable function (PUF)-protected sensor.

**Figure 2 sensors-19-02428-f002:**
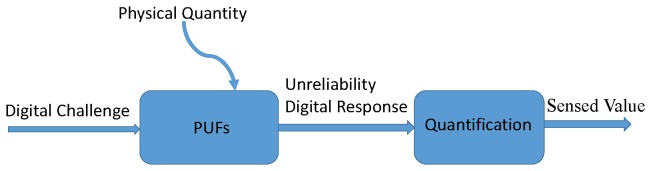
PUF-based sensor.

**Figure 3 sensors-19-02428-f003:**
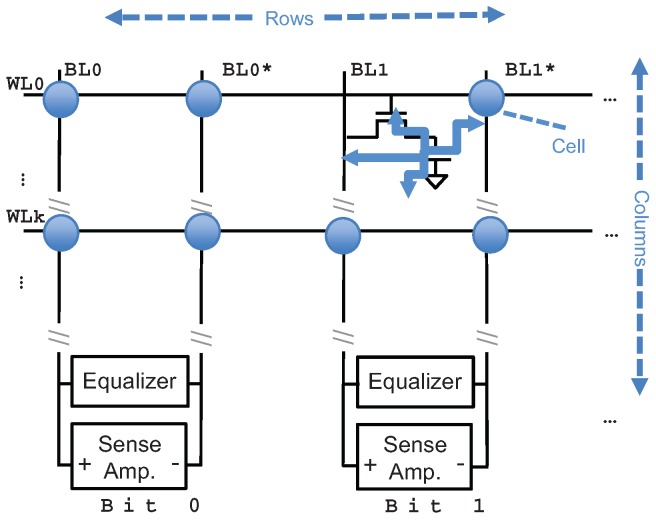
The architecture of DRAM.

**Figure 4 sensors-19-02428-f004:**
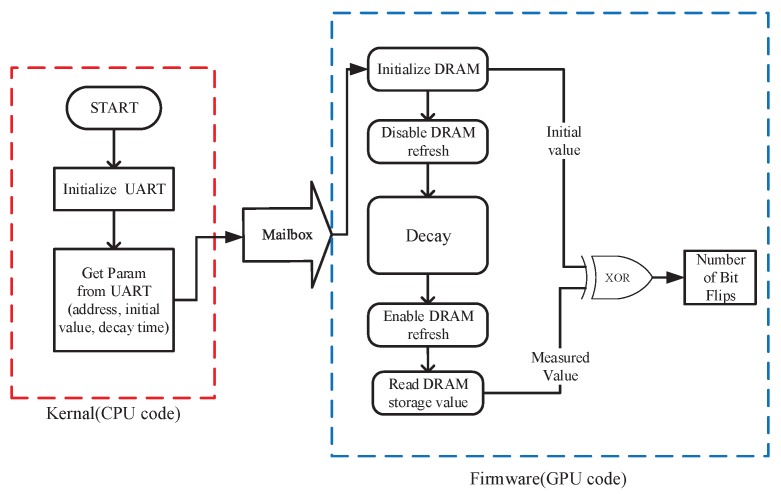
Structure of DRAM PUF implementation on Raspberry Pi (Rpi) B+.

**Figure 5 sensors-19-02428-f005:**
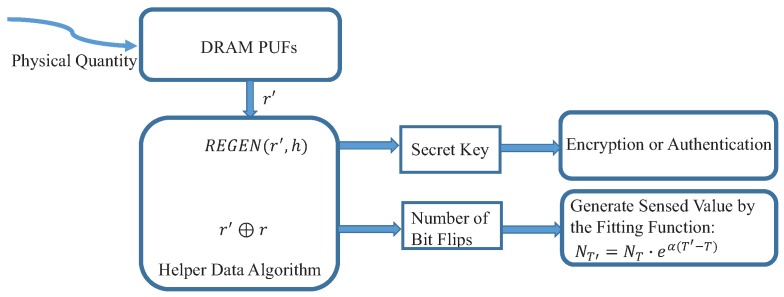
Module of DRAM PUF-based sensor fused with the key generation scheme.

**Figure 6 sensors-19-02428-f006:**
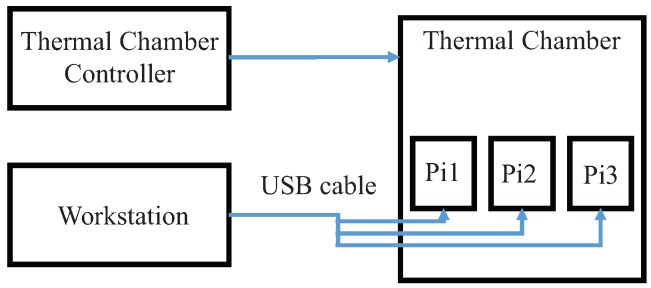
Experimental set up.

**Figure 7 sensors-19-02428-f007:**
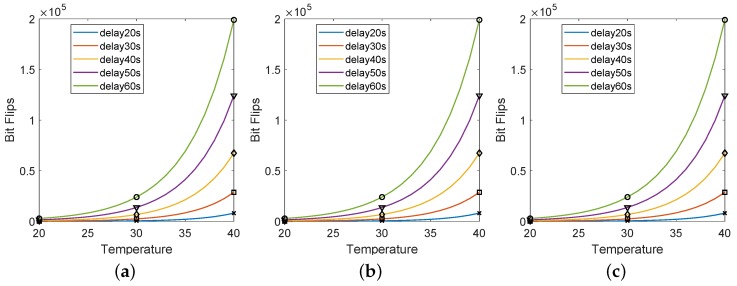
Relation between the temperature, decay time and number of bit flips measured on three Raspberry Pi B+ boards. (**a**) Test results of RPi1. (**b**) Test results of Rpi2. (**c**) Test results of Rpi3.

**Figure 8 sensors-19-02428-f008:**
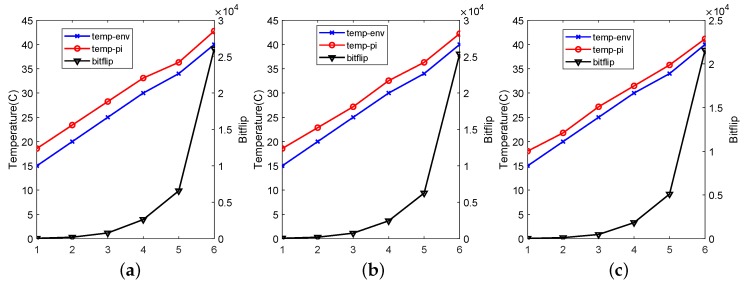
Test results of Rpi1, Rpi2 and Rpi3. The temperature of thermal chamber, temperature sensor on the systems on chip (SOC) and the number of bit flips (right y-ray) are shown in each plot. The x-ray is the number of iteration of tests under different tests conditions. Due to the heat production of function operation, the curve of temperature sensor on the SoC are always higher than the thermal chamber. (**a**) Test results of RPi1. (**b**) Test results of Rpi2. (**c**) Test results of Rpi3.

**Figure 9 sensors-19-02428-f009:**
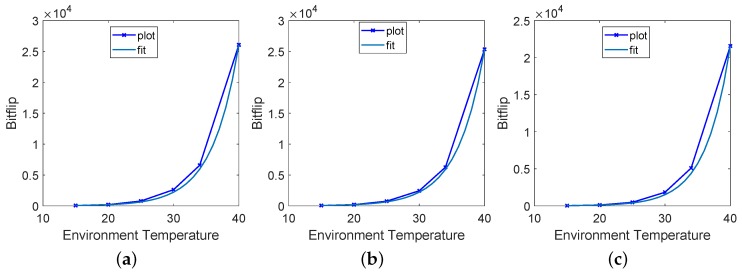
Fitting line of Rpi1, Rpi2 and Rpi3. The smooth curve is the fitting line and the line with the “’X” lable is the original line. (**a**) Fitting line of Rpi1. (**b**) Fitting line of Rpi2. (**c**) Fitting line of Rpi3.

**Figure 10 sensors-19-02428-f010:**
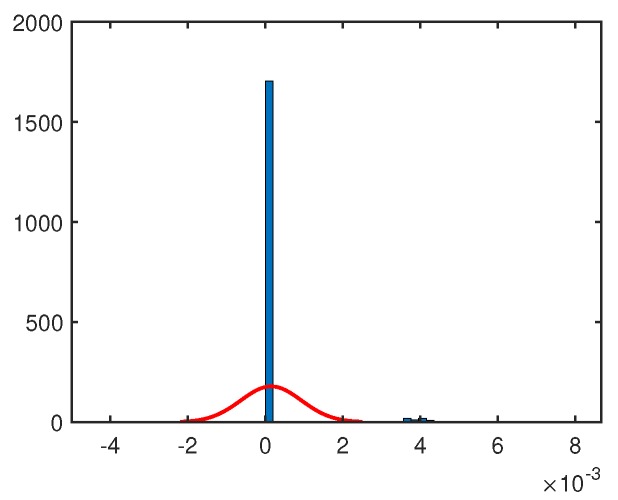
Inter-chip Jaccard index. We evaluate 16MB DRAM as DRAM PUF on each Raspberry Pi board. Given that we do not have enough boards to evaluate the inter-chip Jaccard index, we divided each 16 MB DRAM PUF into 20 parts with the same size. We consider each 0.8MB DRAM PUF as a unique model. Therefore we have 60 DRAM PUF models to operate the Jaccard index evaluation.

**Figure 11 sensors-19-02428-f011:**
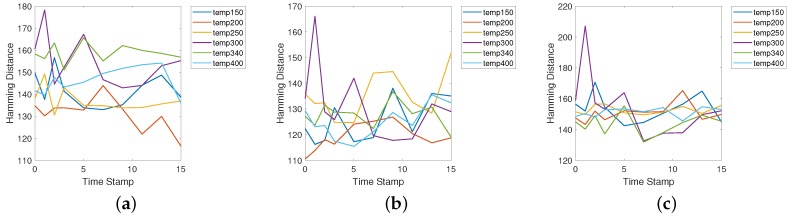
The aging test resuts of Rpi, Rpi2 and Rpi3. The x-ray is the days of the accelerated aging tests under $80{\;}^{\circ }$C. The y-ray is the Hamming distance between the output of DRAM PUF before and after accelerated aging under different temperature. (**a**) Aging test results of Rpi1. (**b**) Aging test results of Rpi2. (**c**) Aging test results of Rpi3.

**Figure 12 sensors-19-02428-f012:**
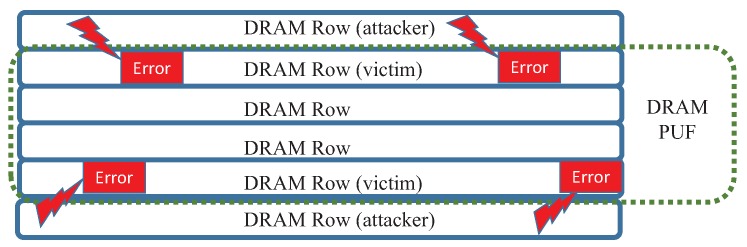
The rowhammer attacks for DRAM PUF.

**Table 1 sensors-19-02428-t001:** The number of bit flips under temperature $20{\;}^{\circ }$C, $25{\;}^{\circ }$C, $30{\;}^{\circ }$C, $35{\;}^{\circ }$C, $40{\;}^{\circ }$C, where TR is the test results in thermal chamber and ER is the value evaluated by Equation (Equation 1).

Rpi	$20{\;}^{\circ }$C		$25{\;}^{\circ }$C		$30{\;}^{\circ }$C		$34{\;}^{\circ }$C		$40{\;}^{\circ }$C	
	TR	ER	TR	ER	TR	ER	TR	ER	TR	ER
Rpi1	198	188	781	647	2621	2219	6571	5948	26074	26103
Rpi2	193	196	751	660	2448	2227	6256	5891	25359	25348
Rpi3	117	106	476	400	1835	1511	5103	4378	21569	21584

**Table 2 sensors-19-02428-t002:** The randomness test results for the DRAM PUF leveraging NIST test suit for temperature 25 ${}^{\circ }$C, 30 ${}^{\circ }$C, 35 ${}^{\circ }$C, 40 ${}^{\circ }$C under decay time 30 s. It should be noticed that as a weak PUF, the length of the output from the DRAM PUF cannot meet some of the tests in the National Institute of Standards and Technology (NIST) test suit, e.g., the length of the bit strings should be longer than $10^{6}$ for the Rank test. Therefore, we just listed the test results that meet the requirement. (N.O.T. is the Non Overlapping Template. FFT is the Fast Fourier Transform test.)

NIST Tests	Pi1				Pi2				Pd i3			
	$25{\;}^{\circ }$C	$30{\;}^{\circ }$C	$34{\;}^{\circ }$C	$40{\;}^{\circ }$C	$25{\;}^{\circ }$C	$30{\;}^{\circ }$C	$34{\;}^{\circ }$C	$40{\;}^{\circ }$C	$25{\;}^{\circ }$C	$30{\;}^{\circ }$C	$34{\;}^{\circ }$C	$40{\;}^{\circ }$C
Frequency	-	-	100%	100%	-	-	97%	97%	-	-	99%	99%
Blcok Frequency	-	-	100%	99%	-	-	100%	97%	-	-	100%	99%
Cumulative Sums	-	-	100%	100%	-	-	99%	96%	-	-	98%	99%
Runs	-	-	100%	99%	-	-	99%	97%	-	-	98%	99%
longest Run	-	-	99%	99%	-	-	0%	98%	-	-	0%	99%
FFT	-	-	95%	98%	-	-	96%	99%	-	-	99%	100%
N.O.T.	-	-	47%	100%	-	-	59%	91%	-	-	98%	98%

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
