# Peer review of "Intrinsic Physical Unclonable Function (PUF) Sensors in Commodity Devices"

_sensors, 2019, doi:10.3390/s19112428_

Round 1

Reviewer 1 Report

In this paper, the authors propose a practical PUF sensor based on DRAM, which is a realizable in RPi, a popular universal device. Since I think this paper contains an interesting and useful ideal for PUF realization, I would like to recommend to accept this paper. 

Further, I think that it will be better to explain the reason of the equation (1).  This formula appears to be related to thermal characteristics of the leakage current in semiconductors. If the theoretical basis can be given, this paper will be even better.

Also, the following minor errors should be corrected before the publication. 

(Minor comments)

1) page 2, line 40, "key or identification are" --> "key or identification is"

2) page 3, line 100, check the citation "[?]" 

3) page 7, at Figure 6, what is the x-axis of these graphs? is it the number of iteration? or what?

4) page 7, line 226, "Figure ??" (please correct the figure number)

5) page 8, line 236, "Figure7" --> "Figure 7"

6) page 9, line 278, please formally express the phrase "can be operated sync with" (what exactly does 'sync' mean here?)

Author Response

We sincerely thank the Editor and Reviewers for their valuable comments and suggestions to enhance the quality of our manuscript. We have addressed all the comments and suggestions given by the reviewers and incorporated it in the revised manuscript. Also, we have submitted a PDF file to provide a point-by-point response to the reviewer's comments.

Reviewer 2 Report

The authors present a novel sensor PUF that relates the number of bit-flips with the temperature.
The main strength of this paper is that they can use off-the-shelf devices without any hardware modification to measure the temperature.
Nevertheless, it is not clear that  this technique can be used in real applications.
The paper must be improved in many different ways (some of them suggested by the authors).
- It is necessary to consider some realistic scenario where the operating conditions change.
The influence of aging, voltage or workload should be addressed at least theoretically.
- Authors should explain the relatation between temperature and time decay.[A]
- The security analysis is poor. Authors should extend this section considering other scenarios (e.g. under/over powering)

There are some typos and missing references in tables and figures.

[A] Xiong, Wenjie et al. “Practical DRAM PUFs in Commodity Devices.” IACR Cryptology ePrint Archive 2016 (2016): 253.

Author Response

(The authors gave the same response as above.)

Reviewer 3 Report

The authors face with the design of a new PUF function.    The design seems novel (DRAM-based approach) and its implementation of a Raspberry Pi is very interesting.  

Unfortunately, the proposal presented in Section 6.1 lacks formal security analysis. 

Two analysis at least are missing: 

1) The randomness of the outputs generated by one DRAM-PUF is missing. Some statistical tests such as the ones provided by DIEHARDER or NIST suits must be used. 

2). An analysis of the haming distance between r and r' is also missing. 

3) A comparative (e.g. correlation) between different PUF chips outputs is missing too.  

In summary, I am not in favour of the acceptance of this article until the authors perform a detailed security analysis. (in the current version Section 6.2 they just wrote three general statements). 

Author Response

(The authors gave the same response as above.)

Round 2

Reviewer 2 Report

The authors have addressed some of my comments. They should consider in future works other scenarios.

Reviewer 3 Report

The authors have addressed, in a minor or less extent, the reviewer's comments.  Therefore I recommend the acceptance of the article. A final proof-reading of the article is recommended.